# TAMING LATENT DIFFUSION MODEL FOR BLACK-BOX MODEL INVERSION ATTACKS

## ABSTRACT

Model inversion attacks (MIAs) seriously threaten privacy by querying models to generate synthetic images that expose features of private training data. These attacks target deep neural networks used in sensitive fields like user authentication. Previous MIA methods based on generative adversarial networks have been proven to be effective, and there are preliminary studies exploring the potential of diffusion models in this field. In this work, we further investigated the feasibility of applying large-scale text to-image latent diffusion models (LDMs) to MIA. To overcome challenges of high training costs from scratch and the complexity of conditional encoding, we propose a novel method that fine-tunes an LDM via an adapter network. Our MI-ControlNet framework integrates probability distributions into the attention module to guide latent space generation. We also train a surrogate model through knowledge transfer and integrated identity loss into the training process of MI-ControlNet, thus establishing a new training paradigm. Experiments on various models and datasets have shown that our method achieves excellent performance in black-box MIAs. Code will be made available following the review process.

## 1 INTRODUCTION

Deep neural networks (DNNs) have been widely applied in multiple key fields, which typically involve highly sensitive private data such as medical images and facial images (Li et al., 2021). Although it is widely believed that the knowledge learned by DNNs from training data is securely encoded in model weights, effectively protecting data privacy, this is not entirely the case. Recent studies have shown that trained models may leak critical information and even unintentionally provide opportunities for privacy attacks by malicious actors (Rigaki & García, 2020; Song & Namiot, 2022). One major threat is model inversion attacks (MIAs), which reverse engineer sensitive training data through model access. For instance, if an attacker successfully attacks a facial recognition model, the user's facial information may be leaked.

Model inversion attacks are classified into white-box (Zhang et al., 2020; Chen et al., 2021; Struppek et al., 2022; Yuan et al., 2023), black-box (An et al., 2022; Han et al., 2023), and label-only attacks (Kahla et al., 2022; Nguyen et al., 2024; Liu et al., 2024). White-box attacks assume that the attacker has full access to the target model's architecture and parameters. Black-box attacks, on the other hand, rely solely on the model's input-output interactions, where attackers infer training data by examining input data and output predictions. Label-only attacks, the most challenging yet applicable in real-world scenarios, grant attackers access only to the model's output labels.

Most existing MIAs leverage generative adversarial networks (GANs) (Radford et al., 2016; Goodfellow et al., 2017), which enhance the effectiveness of attacks by manipulating latent space rather than directly processing pixel space. Recently, LO-CDM (Liu et al., 2024) proposed for the first time a label-only MIA method based on diffusion models (DMs) (Ho et al., 2020), demonstrating the potential application of DMs in the field of MIA. However, this method requires training the DM from scratch and faces the challenge of high computational resources and time costs.

Inspired by (Lin et al., 2023), we have noticed that adapter based fine-tuning methods have successfully applied large-scale text-to-image latent diffusion models (LDMs) (Rombach et al., 2022) in the field of blind image denoising. We have keenly observed that through similar fine-tuning strategies, pre-trained LDMs can be efficiently adapted to MIA tasks. We reformulate the MIA task as a

conditional image generation task, using the target model's output probabilities or predicted labels as conditional signals. The LDM offers inherent advantages: its compact latent space efficiently captures and manipulates sensitive features, while built-in conditioning mechanisms (e.g., cross-attention) enable flexible integration of class-wise information. By fine-tuning a pre-trained LDM via an adapter network (Zhang et al., 2023), only a small number of external parameters are optimized with the backbone fixed, allowing the model to adapt to probability-conditioned generation at low training cost. In contrast to LO-CDM (Liu et al., 2024), which trains a diffusion model from scratch in pixel space, this method efficiently optimizes pre-trained LDM by introducing an adapter network (Zhang et al., 2023), achieving effective alignment between target features and conditions while maintaining model performance.

Regarding the adapter network, ControlNet (Zhang et al., 2023) is a common architecture originally designed for tasks such as image restoration (Lin et al., 2023; Han et al., 2024), where it is conditioned on image input and trained on paired datasets. However, for MIAs, the conditioning information is no longer visual data but rather the model's output labels or class probability distributions. To address this, we propose a novel conditional fusion strategy, termed MI-ControlNet, which embeds probability distributions directly into ControlNet's attention module. Specifically, instead of employing conventional feature concatenation, we embed class probability information into the attention computation, enabling the network to dynamically modulate attention weights based on semantic class guidance. Furthermore, to further improve the effectiveness of the attack, we introduce an identity loss in the MI-ControlNet training process based on a substitute model, thus designing a new training paradigm adapted for MIA tasks. The main contributions of this paper include:

- We are the first to propose an innovative method that leverages an adapter network to control the LDM for the task of MIAs.
- We design MI-ControlNet, which incorporates a novel conditional setting by embedding probability information into the attention module.
- We integrate identity loss based on substitute model into the training process of MI-ControlNet, thereby establishing a novel training paradigm in the black-box setting.
- We demonstrate our method's effectiveness on four models and dataset configurations. Ablation studies confirm the importance of each module, and our approach shows superior performance compared to existing methods in black-box attacks.

## 2 RELATED WORK

MIAs were first introduced by (Fredrikson et al., 2014) for attacking linear regression models and (Fredrikson et al., 2015) proposed a gradient descent-based algorithm to attack shallow networks. Subsequently, MIAs leverage the training of image prior models, such as GANs (Goodfellow et al., 2017), to generate feedback images and optimize the latent vectors. Based on the type of attack, access to the target model can be categorized into white-box, black-box, and label-only MIAs.

**White-Box MIAs.** The work presented in (Zhang et al., 2020) first introduced the use of GANs to address MIAs by optimizing the latent vector fed into the GAN instead of pixel-level optimization, thereby avoiding the generation of meaningless images and laying the foundation for further advancements in the field. (Chen et al., 2021) introduced a specialized GAN training method, where the output of the target model is used to assist in training a knowledge-rich generator during the second stage of the attack. Building on (Chen et al., 2021), (Yuan et al., 2023) proposed the use of conditional GANs (Wang et al., 2018) to replace traditional GANs, thereby avoiding class confusion. Additionally, (Nguyen et al., 2023) suggested that model enhancement techniques, such as ensemble methods, can significantly improve the success rate of attacks. In white-box attacks, both (Wang et al., 2021a) and (Struppek et al., 2022) employed StyleGAN (Karras et al., 2020) for high-resolution inversion; (Wang et al., 2021a) demonstrated that StyleGAN produces more effective attack results compared to traditional DCGANs (Radford et al., 2016), while (Struppek et al., 2022) proposed a novel method for high-resolution model inversion across different data distributions.

**Black-Box MIAs.** The work in (An et al., 2022) introduced a black-box attack method based on StyleGAN, utilizing the classical genetic algorithm (Das & Suganthan, 2011) for optimization, and explored the attack effectiveness in different latent spaces of StyleGAN. RLB-MI (Han et al., 2023)

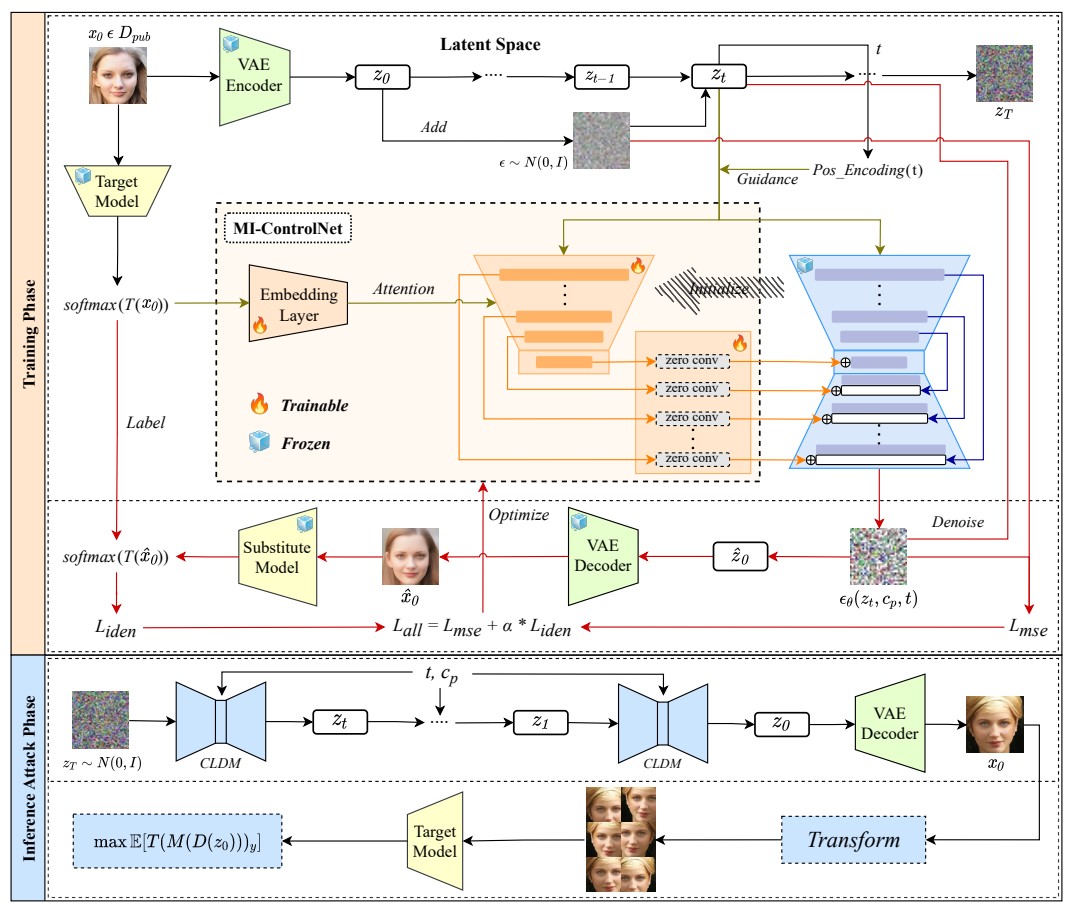

Figure 1: Overview of our proposed method. *CLDM:* Controllable image generation by controlling LDM through MI-ControlNet. The framework incorporates several contributions: 1) We are the first to propose a method that leverages an adapter network to control LDM of black-box MIAs. 2) We introduce MI-ControlNet, which conditions the attention module by embedding class probability information instead of using the concatenation of the condition $c_p$ and the noisy latent $z_t$ at time $t$ as input. 3) We incorporate identity loss based on a substitute model into the training process of MI-ControlNet, thereby establishing a novel training paradigm under the black-box setting.

proposed an optimization method using reinforcement learning, where the loss function is constructed in alignment with the concept of max-margin loss. The method in (Ye et al., 2023) learns the mapping between $y$ and the StyleGAN $w$ space through training, then uses an inversion network to generate seeds optimized by the genetic algorithm (Das & Suganthan, 2011).

**Label-Only MIAs.** In the field of label-only attacks, (Kahla et al., 2022) proposed an algorithm called Boundary Repulsion Model Inversion, which moves toward the target class center by evaluating the model's response to the predicted labels. Subsequently, (Nguyen et al., 2024) introduced a method that trains a substitute model through knowledge transfer, marking the first attack strategy in a label-only setting to adopt the concept of knowledge transfer. LO-CDM (Liu et al., 2024) proposed a method based on DDPM (Ho et al., 2020), where a Diffusion Model is trained by integrating time step embeddings with class label embeddings to conduct MIAs.

## 3 BACKGROUND

**Diffusion Models.** Diffusion models represent a class of generative probabilistic models designed to approximate the underlying data distribution $p(x)$ by iteratively denoising samples perturbed by Gaussian noise. In the realm of image synthesis, the most effective models rely on reweighted variations of the variational lower bound (VLB) (Dhariwal & Nichol, 2021), closely aligned with

denoising score matching techniques (Song et al., 2020). These models can be framed as a series of denoisers, $\epsilon_\theta(x_t, t)$, each trained to predict the clean data from its noisy counterpart $x_t$, where $x_t$ is a perturbed version of the original input $x_0$, with t uniformly sampled from $\{1, 2, 3, \ldots, T-1, T\}$. The relationship between $x_t$ and $x_0$ is given by:

$$x_t = \sqrt{\overline{\alpha_t}}x_0 + \sqrt{1 - \overline{\alpha_t}}\epsilon \tag{1}$$

where $\alpha_t = 1 - \beta_t$, and $\overline{\alpha}_t = \prod_{s=1}^{t} \alpha_s$, $x_0$ represents the original data, and $\epsilon \sim \mathcal{N}(0, 1)$ is Gaussian noise. $\beta_t$ is a parameter that controls the amount of noise added at each time step during the diffusion process, typically chosen to be small for stable training. Incorporating this formulation into the model's objective function leads to the following loss (Ho et al., 2020):

$$L_{dm} = \mathbb{E}_{x_t, \epsilon, t} \left[ \|\epsilon - \epsilon_\theta(x_t, t)\|^2 \right] \tag{2}$$

This objective facilitates the model's capacity to recover the clean data through iterative denoising.

**Class-Conditional Diffusion Models.** Conditional diffusion models can be categorized into two types: classifier-guidance (Dhariwal & Nichol, 2021) and classifier-free guidance (Ho & Salimans, 2022). LO-CDM (Liu et al., 2024) falls under the classifier-free category and is built upon DDPM (Ho et al., 2020), where the diffusion model is trained by fusing the time step embedding and the class label embedding. The loss function is modified as follows (Liu et al., 2024):

$$\mathcal{L}_{dm} = \mathbb{E}_{x_t, \epsilon, t, y} \left[ \|\epsilon - \epsilon_\theta(x_t, t + y)\|^2 \right] \tag{3}$$

Here, the denoiser $\epsilon_\theta$ not only relies on the noisy input $x_t$ and time step $t$, but also on the class label $y$. By embedding class information into the time step $t$, the model gains the ability to steer the generation towards specific class-conditioned features, thereby achieving more accurate control over the class attributes during image generation.

# 4 METHODOLOGY

In this section, we systematically introduce the method proposed in this paper, namely the **C**ontrol **L**atent **D**iffusion Model for **M**odel **I**nversion Attacks (CLD-MI).

## 4.1 PROBLEM FORMULATION

**Attacker's Goal.** The objective of MIAs is to reconstruct representative data for a specific class $y$ from a target model $T$, which has been trained on a private dataset. The target model $T : x \to [0, 1]^K$ maps an input image $x \in \mathbb{R}^d$ to a label, where $K$ represents the number of classes.

**Attacker's Knowledge.** In the black-box setting, the attacker can only query the model using input data of their choice and obtain the corresponding soft labels as output. Additionally, the attacker has knowledge of the target model's purpose. Previous researchs (Struppek et al., 2022; Han et al., 2023; Liu et al., 2024) have shown that information about the task performed by the model or service is typically available or can be inferred from the output classes. Leveraging this task knowledge, the attacker can also access relevant public datasets for the given task.

**Method Overview.** Our method consists of two phases overall. ***Training Phase:*** Public data is first encoded by the VAE encoder $\mathcal{E}$, producing a latent representation $z_0$. Gaussian noise is then added to $z_0$, resulting in a noisy latent representation $z_t$, which is input into both MI-ControlNet and the U-Net (Rombach et al., 2022). MI-ControlNet receives conditional inputs from the target model's output, processed through an embedding layer, and uses these to guide its attention module. Meanwhile, the U-Net predicts the added noise, optimized using MSE loss to ensure image quality. Additionally, an identity loss based on the substitute model is introduced to ensure that the generated images accurately match the target class, enhancing the overall effectiveness of this method. Here, we freeze the U-Net parameters but keep trainable the parameters of both MI-ControlNet and the conditional encoding layer. ***Inference Attack Phase:*** A one-hot vector representing the attack class as the condition to generate the attack results. Following the strategy in (Struppek et al., 2022), we adopt a selection transformation method, to select the output with the highest confidence score from the generated results. Our method is summarised in Figure 1.

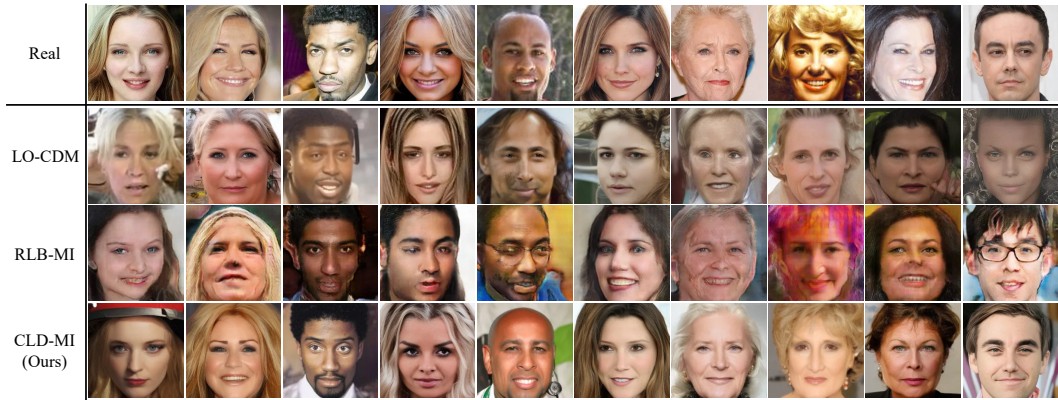

Figure 2: Visualization of the attack with $D_{pub}$ = FFHQ, $D_{pri}$ = CelebA, $T$ = ResNet-18, and $S$ = ResNet-34. The left column represents different methods, while the bottom row illustrates the results obtained by our proposed method.

## 4.2 LATENT DIFFUSION MODEL

Our approach builds upon the large-scale pretrained text-to-image LDM. To ensure higher efficiency and stable training, LDM pretrains an autoencoder that encodes an image $x$ into its latent space representation $z$ via the encoder $\mathcal{E}$, and subsequently reconstructs the image through the decoder $\mathcal{D}$. Both the diffusion and denoising processes occur within the latent space. During diffusion, Gaussian noise with variance $\beta_t \in (0, 1)$ is added to the latent variable $z = \mathcal{E}(x)$ at time step $t$, yielding a noisy latent representation. This process is governed by the following equation:

$$z_t = \sqrt{\bar{\alpha}_t} z_0 + \sqrt{1 - \bar{\alpha}_t} \epsilon \tag{4}$$

where $\epsilon \sim \mathcal{N}(0, I)$, $\alpha_t = 1 - \beta_t$, and $\bar{\alpha}_t = \prod_{s=1}^{t} \alpha_s$. As $t$ increases, the latent variable $z_t$ gradually approaches a standard Gaussian distribution. The network $\epsilon_\theta$ is trained to predict the noise $\epsilon$ at a randomly chosen time step $t$, attention conditioned on $c$ (e.g., text prompts). The objective function for optimizing the latent diffusion model is formulated as:

$$\mathcal{L}_{\text{ldm}} = \mathbb{E}_{z_t, c, t, \epsilon} \left[ \|\epsilon - \epsilon_\theta(z_t, c, t)\|^2 \right] \tag{5}$$

where $z = \mathcal{E}(x)$, $x$ and $c$ come from the dataset, $t$ is randomly sampled from a uniform distribution, and $\epsilon$ is drawn from a standard normal distribution.

## 4.3 TAMING LDM FOR MIAS

In MIAs, (Chen et al., 2021; Han et al., 2023) marks the training data of the target model $T$ as private data ($D_{pri}$) and the data for training the prior model as public data ($D_{pub}$). Given public data samples $x \sim D_{pub}$, we obtain target model predictions $y = T(x)$ and leverage a pre-trained LDM as the image prior. The process mainly consists of the following three key aspects:

**Conditional Encoding.** We encode the condition as $p = \text{Softmax}(T(x))$. To enhance the control effect of the condition, $p$ is projected through an embedding layer to yield a latent condition representation $c_p = \text{Embedding}(p)$. This representation is then integrated into MI-ControlNet to guide generation, with the embedding layer treated as a learnable component. Unlike (Zhang et al., 2023; Lin et al., 2023), the condition is applied within the attention module rather than being concatenated with $z_t$ (noise-added $z_0$) as input to ControlNet. In traditional ControlNet, the condition is typically an image (e.g., using a low-quality image to fine-tune the LDM for high-quality image reconstruction). But in MIA scenarios, our condition is class information, which is more effectively applied within the attention module. We named this network MI-ControlNet.

**MI-ControlNet.** Following the ControlNet (Zhang et al., 2023) and construct a trainable copy of the pre-trained U-Net encoder and intermediate block, denoted as $M_{\text{cond}}$. MI-ControlNet is composed of $M_{\text{cond}}$, zero convolution layers, and an embedding layer. The intermediate block receives conditional information and outputs control signals. By directly copying the weights instead of random

| $D_{pub} \rightarrow D_{pri}$ | Method | Acc@1 ↑ | Acc@5 ↑ | $\delta_{face}$ ↓ | FID ↓ |
|---|---|---|---|---|---|
| | LO-CDM | 31.24% | 62.31% | 0.9013 | 79.28 |
| CelebA → CelebA | RLB-MI | 72.32% | 88.36% | 0.7429 | 59.65 |
| | CLD-MI | **88.73%** | **98.29%** | **0.6303** | **49.76** |
| | LO-CDM | 24.35% | 51.29% | 0.8625 | 84.35 |
| FFHQ → CelebA | RLB-MI | 41.28% | 74.28% | 0.8231 | 79.69 |
| | CLD-MI | **78.23%** | **93.22%** | **0.7044** | **68.67** |
| | LO-CDM | 26.32% | 51.25% | 0.8312 | 89.32 |
| FFHQ → FaceScrub | RLB-MI | 40.58% | 71.26% | 0.8013 | 86.35 |
| | CLD-MI | **93.86%** | **99.61%** | **0.7068** | **73.26** |
| | LO-CDM | 59.32% | 75.32% | 0.7869 | 70.23 |
| CelebA → FaceScrub | RLB-MI | 73.25% | 91.28% | 0.7536 | 65.21 |
| | CLD-MI | **96.81%** | **100.00%** | **0.6962** | **50.98** |

Table 1: Comparison of different MIA methods using $T$ = ResNet-18 trained on various datasets, with $S$ = ResNet-34. CLD-MI (Ours) outperforms other methods in all metrics.

| $D_{pub} \rightarrow D_{pri}$ | Architecture | Acc@1 ↑ | Acc@5 ↑ | $\delta_{face}$ ↓ | FID ↓ |
|---|---|---|---|---|---|
| | ResNeSt-101 | **87.71%** | 97.27% | **0.6234** | 50.45 |
| CelebA → CelebA | ResNet-152 | 85.83% | **98.07%** | 0.6337 | 51.18 |
| | DenseNet-169 | 86.80% | 97.20% | 0.6588 | **50.38** |
| | ResNeSt-101 | 77.23% | **92.62%** | **0.7178** | **71.45** |
| FFHQ → CelebA | ResNet-152 | 76.88% | 89.29% | 0.7354 | 73.67 |
| | DenseNet-169 | **78.01%** | 90.84% | 0.7377 | 72.16 |
| | ResNeSt-101 | **93.26%** | **99.73%** | 0.7145 | 74.75 |
| FFHQ → FaceScrub | ResNet-152 | 92.21% | 99.41% | **0.7016** | 75.59 |
| | DenseNet-169 | 93.02% | 99.60% | 0.7223 | **74.32** |
| | ResNeSt-101 | **95.22%** | **100.00%** | 0.7288 | **49.71** |
| CelebA → FaceScrub | ResNet-152 | 94.43% | 99.20% | **0.7367** | 50.21 |
| | DenseNet-169 | 94.20% | 99.42% | 0.7564 | 51.14 |

Table 2: Comparison of architectures across different settings with Acc@1, Acc@5, $\delta_{face}$, and FID.

initialization, we provide a strong weight initialization for $M_{cond}$. Additionally, the first layer of $M_{cond}$ is initialized to zero to prevent gradient vanishing caused by random noise during the early stages of training (Lin et al., 2023). For the conditional design of the attention module, we input $c_p$, which encodes the class probability corresponding to the image. Diverging from conventional ControlNet's multi-scale feature modulation of frozen U-Net denoiser, MI-ControlNet implements zero-initialized convolutions as exclusive skip connections to the U-Net denoiser, ensuring superior training stability. During training, all parameters of MI-ControlNet are updatable. Specifically, we aim to minimize the following latent diffusion objective function:

$$\mathcal{L}_{mse} = \mathbb{E}_{z_t, c_p, t} \left[ \| \epsilon - \epsilon_\theta(\sqrt{\bar{\alpha}_t} z_0 + \sqrt{1 - \bar{\alpha}_t} \epsilon, c_p, t) \|^2 \right] \tag{6}$$

Here, $c_p$ is the class probability condition input to the attention module.

**Identity Loss.** To accelerate the MI-ControlNet's convergence and enhance attack effectiveness, we introduce an identity loss. Obtain the clean latent vector $\hat{z}_0$ from the predicted noise $\epsilon_t$ at time $t$:

$$\epsilon_t = \epsilon_\theta(z_t, c_p, t), \quad \hat{z}_0 = \frac{z_t - \sqrt{1 - \bar{\alpha}_t} \epsilon_t}{\sqrt{\bar{\alpha}_t}} \tag{7}$$

In this step, we aim to guide $\mathcal{D}(\hat{z}_0)$ to learn in the direction that maximizes the probability of outputting the target class $y$. However, since the target model's weights are inaccessible in black-box scenarios, we transfer knowledge from the target model to the substitute model using public data to solve the gradient propagation problem (Nguyen et al., 2024). Hence, we define the identity loss based on the substitute model as follows:

$$\mathcal{L}_{iden} = \mathcal{L}_{MM}(S(\mathcal{D}(\hat{z}_0)), y) \tag{8}$$

where $MM$ denotes the max-margin loss (Yuan et al., 2023), $S$ represents the substitute model. $MM$ loss is defined as:

$$\mathcal{L}_{MM} = -p_y(x) + \max_{j \neq y} p_j(x) \tag{9}$$

where $p_y(x)$ denotes the confidence score for target class $y$. $\max_{j \neq y} p_j(x)$ denotes the maximum confidence score for all classes except target class. $K$ is the total number of classes. Complete loss for training MI-ControlNet is given by:

$$\mathcal{L}_{all} = \mathcal{L}_{mse} + \alpha \, \mathcal{L}_{iden} \tag{10}$$

where $\alpha$ is the weight factor, set to 0.025. After training MI-ControlNet, we set the corresponding one-hot vector for class $y$. Then, we obtain the corresponding conditions $c_p$ through the embedding layer and input these conditions into MI-ControlNet to achieve the desired inference results.

| $D_{\text{pub}} \to D_{\text{pri}}$ | Architecture | Acc@1 ↑ | Acc@5 ↑ | $\delta_{\text{face}}$ ↓ | FID ↓ |
|---|---|---|---|---|---|
| | ResNet-34 | **96.81%** | **100.00%** | **0.6962** | **50.98** |
| CelebA → FaceScrub | EfficientNet-B2 | 92.49% | 99.41% | 0.7209 | 52.41 |
| | DenseNet-161 | 93.27% | 99.83% | 0.7307 | 51.44 |

Table 3: Ablation study results under different substitute model architectures, with $T$ = ResNet-18.

| $D_{\text{pub}} \to D_{\text{pri}}$ | Loss | Setting | Acc@1 ↑ | Acc@5 ↑ | $\delta_{\text{face}}$ ↓ | FID ↓ |
|---|---|---|---|---|---|---|
| | $\mathcal{L}_{CE}$ | w/ Prob Emb | 62.45% | 78.38% | 0.8518 | 62.12 |
| | | w/o Emb | 00.13% | 00.31% | 1.3381 | 67.98 |
| CelebA → CelebA | $\mathcal{L}_{MM}$ | w/ Label Emb | 76.62% | 94.37% | 0.7212 | 58.02 |
| | | w/o Selection Strategy | 75.75% | 95.55% | 0.6791 | 55.73 |
| | | w/ Prob Emb | **85.83%** | **98.07%** | **0.6337** | **51.18** |

Table 4: Ablation study results across three modules: the conditional module, identity loss, and selection strategy, with $T$ = ResNet-152 and $S$ = ResNet-34.

### 4.4 Selection Strategy

After obtaining multiple attack results from multi-round inference, we aim to select the images that best reflect the characteristics of the target model's $D_{pri}$. Following previous work (Struppek et al., 2022; Liu et al., 2024), we apply a transformation selection strategy to the inferred images to compute the expected confidence scores after transformation, as detailed in the following formula:

$$E\left[T(M(\mathcal{D}(z)))_y\right] \approx \frac{1}{N} \sum_{i=1}^{N} T(M(\mathcal{D}(z)))_y \tag{11}$$

For the calculation of expectations, we employ a Monte Carlo method for estimation, where $M$ represents the transformation operation.

## 5 Experiments

### 5.1 Experimental Setting

**Datasets.** We evaluate our method on classifiers trained with two representative face datasets: CelebA (Liu et al., 2015) and FaceScrub (Ng & Winkler, 2014). Following previous work (Struppek et al., 2022), CelebA was divided into private and public datasets, with the former used to train the target and evaluation models, and the latter for surrogate model and MI-ControlNet training. To evaluate the performance of our method under the distribution shift scenario between private and public datasets, we conducted experiments using the Flickr Faces HQ Dataset (FFHQ) (Karras et al., 2019) as the public dataset. Furthermore, we use CelebA as public dataset to attack models trained on Facescrub to further simulate distribution shift scenario. The private and public datasets are completely disjoint. For more information on the datasets, please refer to the Appendix.

**Models.** We conduct our attacks on several widely used network architectures to ensure fair comparisons. Following previous work (Struppek et al., 2022), we experiment with four target model architectures: ResNet-18, ResNet-152 (He et al., 2016), ResNeSt-101 (Zhang et al., 2022) and DenseNet-169 (Huang et al., 2017). The evaluation model adopts Inception-V3 (Szegedy et al., 2016). We tried various substitute model, including ResNet-34 (He et al., 2016), EfficientNet-B2 (Tan & Le, 2019), and DenseNet-161 (Huang et al., 2017), and ultimately chose ResNet-34.

MI-ControlNet is built on the ControlNet (Zhang et al., 2023), integrating embedding layer and zero convolutional layers. Compared with the original ControlNet architecture, we achieved architecture simplification by reducing the number of residual blocks at each resolution level. The main architecture parameters of MI-ControlNet were initialized with pre-trained encoder parameters of Stable Diffusion v2.1 (Rombach et al., 2022). Meanwhile, the zero convolutional layers adopts a zero weight initialization strategy to ensure training stability (Lin et al., 2023), while the embedding layer was implemented using PyTorch's standard embedding function and randomly initialized with parameters. More details for both the classifiers and MI-ControlNet are provided in Appendix.

**Evaluation Metrics.** Following previous work (Struppek et al., 2022; Han et al., 2023), we used several evaluation metrics to assess the performance of our method. ***Attack Accuracy (Acc@1,***

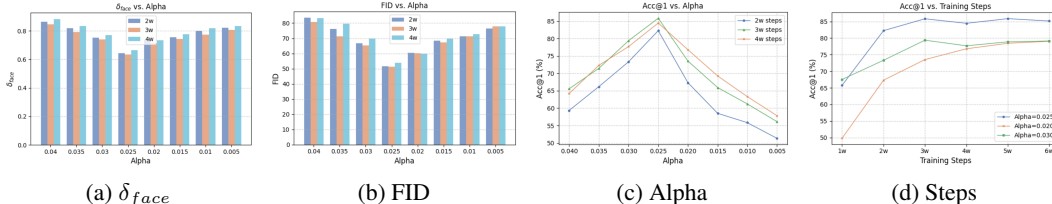

| (a) $\delta_{face}$ | (b) FID | (c) Alpha | (d) Steps |

Figure 3: The figure illustrates: (a) the variation of $\delta_{face}$ with respect to different $\alpha$ values, (b) the changes in FID as $\alpha$ varies, (c) the attack success rate across varying $\alpha$ values, and (d) the relationship between accuracy and the number of steps.

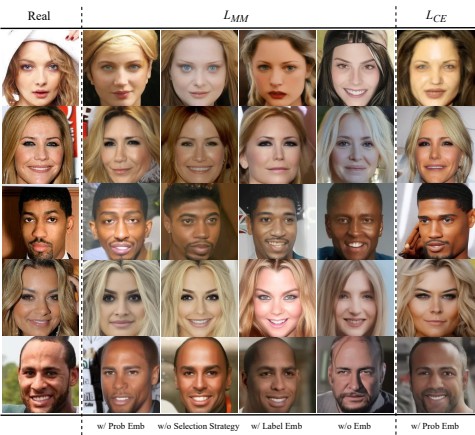

Figure 4: The visualization of ablation results. The leftmost column displays ground truth images, with subsequent columns showing ablation results under different settings.

***Acc@5):*** The evaluation model inception-v3, trained on public datasets was used to predict the labels of attack results generated by target class conditional guidance and report Top-1 and Top-5 attack accuracy. ***Mean Shortest Feature Distance ($\delta_{face}$):*** Computed the shortest feature distance between the generated image of each target class and any training data in that class, with the average distance denoted as $\delta_{face}$. These distances were measured by calculating the $l_2$ norm squared of the activation in the penultimate layer of a pre-trained FaceNet model (Schroff et al., 2015). Lower values indicate attack results that are visually closer to the training data. ***Fréchet Inception Distance (FID)*** (Heusel et al., 2017)*:* FID quantifies feature-space similarity between target training data and attack results. The feature vectors were extracted using an inception-v3 model trained on ImageNet (Deng et al., 2009). A lower FID score indicates greater similarity between the two datasets.

## 5.2 EXPERIMENTAL RESULTS

**Comparison with Previous MIA Methods.** We conducted a comprehensive performance comparison of our method with RLB-MI (Han et al., 2023) and LO-CDM (Liu et al., 2024). RLB-MI is a black-box MIA method based on reinforcement learning, with an original attack resolution of $64 \times 64$. Following the methodology outlined in (Struppek et al., 2022), we upgraded the GAN architecture of RLB-MI to support a higher resolution of $256 \times 256$ for fair comparison. LO-CDM is the first diffusion model-based method for the label-only scenario. Since the LO-CDM's code is not publicly available, we strictly followed the method proposed in the paper for implementation. Specifically, we utilised the $D_{pub}$ and incorporated class information at each timestep to train a DDPM model(Ho et al., 2020). To ensure fair comparison, we incorporate probability output rather than class label at each time step. It is worth noting that training a DDPM using the LO-CDM method is estimated to take approximately one week, whereas our method requires only 16 hours. Please refer to the Appendix for a detailed comparison of resource consumption.

We present a comparative analysis of various MIA methods across four data distribution settings. Our method consistently outperforms RLB-MI and LO-CDM across all metrics (Table 1), indicat-

ing its superior performance in terms of reconstruction accuracy and visual quality. These results suggest that CLD-MI effectively adapts to diverse data distributions, leading to more reliable and realistic outputs. The visualization results are shown in Figure 2, where we provide real samples alongside the attack images generated by the MIA methods. It can be seen that our method performs excellently in terms of fidelity.

**Extended Evaluation.** Table 2 shows the evaluation results of different datasets and target model architectures. The results show that our method achieves the highest Acc@1 across most settings on ResNeSt-101, especially reaching 95.22% in Celeba $\rightarrow$ Facescrub, Meanwhile, our method also demonstrates competitive performance on other architectures, but exhibits volatility in $\delta_{face}$ and FID. In FFHQ $\rightarrow$ CelebA, our method's performance decreased across all architectures, especially for ResNet-152, where Acc@1 dropped to 76.88%. Since the large number of target class (1000) in $D_{pri}^{CelebA}$ and the number of $D_{pub}^{FFHQ}$ is small, the attack model converges slows down, resulting in a decrease in accuracy. In contrast, $D_{pri}^{FaceScrub}$ has only 530 classes, making the number of $D_{pub}^{FFHQ}$ relatively sufficient, thereby achieving better performance.

## 5.3 ABLATION STUDY

We conducted comprehensive ablation experiments on our method (CLD-MI) to assess the impact of the following four key factors on the attack performance of the method.

**Substitute Model Architecture Selection.** We adopt the knowledge transfer strategy to train a white-box surrogate model to solve the issue of gradient backpropagation failure during MI-ControlNet training caused by the inaccessibility of target model parameters in the black-box scenario. As shown in Table 3, the choice of surrogate model architecture have no substantial impact on the attack efficacy, with ResNet-34 exhibiting a slight performance improvement.

**Importance of Condition Module.** We evaluated the impact of different condition settings on the performance of our method, with $\alpha = 0.025$ and steps = 3w. The results in Table 4 show that after embedding the hard labels, Acc@1 can reach 76.62%, while embedding the probabilities can increase Acc@1 to 85.83%. In contrast, without any embedding, Acc@1 drops to 0.1%, indicating that the condition module is crucial.

**Analysis of $\mathcal{L}_{iden}$ and Selection Strategy.** We conducted a series of ablation studies to evaluate the choice of $\mathcal{L}_{iden}$. Previous work has highlighted the gradient vanishing issue associated with cross-entropy loss ($\mathcal{L}_{CE}$) (Struppek et al., 2022; Yuan et al., 2023). When $\mathcal{L}_{iden} = \mathcal{L}_{CE}$, the attack results can be found in Table 4. To mitigate gradient vanishing while training the MI-ControlNet, we adopted $\mathcal{L}_{MM}$ proposed by (Yuan et al., 2023). Table 4 also demonstrates the effectiveness of the selection strategy, resulting in an approximate 10% improvement in Acc@1.

**Effect of the Alpha and Steps.** We explored the relationship between Acc@1 and $\alpha$. As shown in Figure 3, when $\alpha$ exceeds 0.025, the weight of $\mathcal{L}_{iden}$ becomes too large, causing significant fluctuations in $\mathcal{L}_{mse}$ and generating non-meaningful images. Conversely, setting $\alpha$ too low slows down convergence, reducing the effectiveness of the attack. Through extensive testing, we determined $\alpha$ to be 0.025. With $\alpha$ fixed at 0.025, further analysis of the relationship between Acc@1 and steps showed that the success rate stabilizes around 30,000 steps, with minimal improvements beyond this point. In addition to quantitative data, the visualisation results of ablation studies (Figure 4; Figure 7 and 8 in Appendix) further confirm the impact of key factors.

## 6 CONCLUSION

This paper proposes MI-ControlNet, a novel framework that enhances MIA effectiveness by embedding class probability distributions into the attention mechanism. In addition, we introduce $\mathcal{L}_{iden}$ based on substitute model into the training process, creating a specialized training paradigm for MIA. The experimental results show that our method achieves superior performance in black-box scenarios across various settings. These findings not only prove the effectiveness of our method, but also highlight the potential of LDM in conducting complex privacy attacks. This study provides a promising avenue for future exploration of the vulnerability of neural networks and advancement of MIA technology.

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
