# OpenReview forum: "Taming Latent Diffusion Model for Black-Box Model Inversion Attacks"
_ICLR.cc/2026/Conference — ICLR 2026 Conference Withdrawn Submission_

### Official Review · Reviewer_W1Wu · 2025-10-28

**Soundness:** 2
**Presentation:** 3
**Contribution:** 2
**Rating:** 4
**Confidence:** 4

**Summary:**

This paper proposes a new method for black-box model inversion attacks, named CLD-MI. This method utilizes a powerful pre-trained latent diffusion model as the generator. To effectively control the LDM, the authors designed an adapter network called MI-ControlNet. This network takes the class probability vectors obtained from the target model as a condition, injecting them directly into the attention module to guide the LDM's image generation.

The method utilizes a substitute model trained on a public dataset to calculate an identity loss, which forces the generated images to accurately reflect the features of the target class. Experimental results show that the proposed method achieves higher attack accuracy and superior image quality compared to existing black-box and label-only MIA methods.

**Strengths:**

1. The paper is well-organized, and the proposed method is presented clearly.

2. The authors conducted thorough experiments across a variety of settings, including different datasets and target model architectures.

3. The paper proposes a well-designed framework that effectively utilizes LDM and ControlNet to generate high-quality attack images.

4. The proposed method achieves state-of-the-art results across multiple standard evaluation metrics.

**Weaknesses:**

1. Lack of analysis on query complexity: While older attacks focused on succeeding at all, modern black-box attacks must be query-efficient to avoid detection and minimize cost. The proposed method requires querying the target model to obtain probability vectors for training its MI-ControlNet, yet the paper provides no quantification of the total queries needed for this training stage. This omission makes it difficult to assess the attack's practicality in real-world, query-limited scenarios.

2. The comparison to the RLB-MI baseline seems unfair. The new method (CLD-MI) utilizes a powerful, pre-trained diffusion model (LDM) that was trained on a massive dataset, giving it significant prior knowledge. In contrast, the baseline's GAN was trained from scratch using only the smaller, experiment-specific public dataset. This major difference in pre-training is a significant confounding variable. The reported performance gains might stem from this superior prior knowledge rather than the novel components of MI-ControlNet. While using a pre-trained model is practical for a real-world attack, a fairer methodological comparison would require training the diffusion model from scratch on the same public data used by the baseline.

3. Minor points

- The framework, while effective, appears to be a well-engineered combination of existing techniques (ControlNet, identity loss, selection transformation) rather than introducing a fundamentally new component. The novelty lies primarily in the application and integration of these parts for the MIA task. While valuable, this feels somewhat incremental from a methodological standpoint, making it questionable whether the work has sufficient impact for a top-tier venue like ICLR.

- The paper incorrectly uses the term loss function (L145-146) when referring to the objective of the RLB-MI method. Since it uses reinforcement learning, this should be termed a reward function.

**Questions:**

1. Could the authors provide an analysis of the query complexity? Specifically, how many queries to the target model are required during the training phase, and how does this impact the attack's practicality in real-world, query-limited scenarios?

2. Have the authors considered a fairer comparison by training the LDM from scratch on the same public data?

---

### Official Review · Reviewer_V1m4 · 2025-10-28

**Soundness:** 3
**Presentation:** 2
**Contribution:** 2
**Rating:** 2
**Confidence:** 4

**Summary:**

This paper proposes MI-ControlNet, a black-box model inversion attack framework based on the Latent Diffusion Model (LDM). The method embeds class probability distributions into the attention modules of a pretrained LDM to achieve probability-guided latent-space generation, while incorporating a substitute model and an identity loss to enable trainable optimization under black-box conditions. Extensive experiments on CelebA, FFHQ, and FaceScrub demonstrate that MI-ControlNet significantly outperforms RLB-MI and LO-CDM in terms of attack accuracy (Acc@1) and generation quality (FID). The main contributions include: (1) adapting pretrained LDMs to black-box MIA tasks via adapter-based fine-tuning; (2) designing a probability-embedded conditional attention mechanism; and (3) introducing a surrogate-based identity loss to form a new training paradigm.

**Strengths:**

Practical significance of the problem:
The paper targets privacy risks posed by generative models and explores how latent diffusion models (LDMs) can be exploited for black-box inversion attacks. The problem is well-defined and closely aligned with real-world threats.

Rational and efficient method design:
The authors employ adapter-based fine-tuning instead of training a diffusion model from scratch, which substantially reduces training costs. By freezing the LDM backbone and optimizing only the control module and embedding layer, the approach achieves stable training and manageable computational overhead.

Effective probability embedding strategy:
Embedding class probability distributions directly into the attention module, instead of concatenating them with the input, improves class consistency in generated images. This design is empirically validated and performs better than label embedding.

Comprehensive experiments:
The paper evaluates performance across multiple dataset-transfer settings (CelebA, FFHQ, FaceScrub) and includes ablation studies (α parameter, training steps, embedding strategy, substitute models), demonstrating consistent results.

Engineering relevance:
MI-ControlNet shows a feasible direction for task adaptation without impairing the generative capability of pretrained diffusion models, providing valuable insights for future research on privacy attacks and defenses.

**Weaknesses:**

Limited theoretical innovation
MI-ControlNet largely follows the existing ControlNet architecture, modifying the input condition from an image to a class probability distribution embedded into the attention module. Although this yields empirical improvements, the paper lacks new modeling insights or mechanistic analysis. It does not explain why probability embedding improves inversion performance or how it influences attention distributions or latent-space generation dynamics. Overall, the contribution is more of an empirical structural modification than a conceptual advance.

Overly relaxed black-box assumption
While claiming to address black-box model inversion, the experiments assume that attackers can access the target model’s full probability outputs (soft labels). This assumption rarely holds in practical scenarios, as most APIs or commercial models only return discrete labels or confidence scores. If only top-1 labels were available, the proposed probability embedding and surrogate training mechanism would not apply. The paper does not evaluate the method under such constrained conditions, making the attack setting somewhat idealized.

Narrow experimental scope
Although MI-ControlNet is theoretically applicable to general black-box inversion tasks, the experiments are conducted exclusively on face recognition models and evaluated using identity-similarity metrics. This choice is representative within the MIA literature but insufficient to demonstrate broader applicability across model types or modalities. The authors are encouraged to include non-face or cross-modal experiments in future work, or to explicitly state that the current study focuses on face recognition as the primary evaluation scenario, thereby aligning the experimental scope with the paper’s claims.

Insufficient theoretical explanation
While the paper contains a reasonable number of equations, most restate standard diffusion model formulations rather than deriving novel mechanisms. The proposed “probability-embedded attention” and “surrogate-based identity loss” are presented empirically without mathematical justification showing how they modify attention distributions or gradient propagation. Consequently, the theoretical rationale for the method’s effectiveness remains underdeveloped.

Additionally, the writing and formatting could be improved. All experimental tables (Table 1–4) underutilize column width, resulting in sparse layouts and unnecessary whitespace.

**Questions:**

See Weaknesses

---

### Official Review · Reviewer_AkJd · 2025-10-30

**Soundness:** 2
**Presentation:** 2
**Contribution:** 2
**Rating:** 4
**Confidence:** 2

**Summary:**

This work use text-to-image LDM in the pipeline of model inversion attack i. e. trying to reconstruct data for a specific class used to train the target model. The authors present a modification of previous work called MI-ControlNet that given the probability distributions of the target model incorporates this information to guide the latent space generation of DM. They also distill black-box model into a surrogate model to add a specific identity loss into the training of MI-controlNet.

**Strengths:**

The authors show how to fit the pre-trained LDM into MIA pipeline. The results seems to confirm the successfulness of the proposed method. Ablation studies are reported for the proposed approach. The work may be of interest for people working on MIAs.

**Weaknesses:**

I may not be the best fit to review this work, I can only comment on some general issues and what is not clear for me:

W1.

Threat model issues: I may not be familiar with a standard MIA setup, but:
- is the adversary is not restricted and can use any data? Its it a common assumption for MIAs? D_pub should be formally introduced in problem formulation but it is not mentioned and described later on.
- Distilling a black-box model into a surrogate model with the full logits returned without any query number restriction is raising my concerns.
- are all proposed MIA attacks high-resource consuming attacks? Seems unlikely given assumption the adversary can use any public data.

W2.

Baselines choice:
- 1) results for validation data of target model on real data are needed to see the effectiveness of the method.
- 2) two compared baselines looks like a weak comparison, since the authors changed baselines they compare to (resizing images for one using GANs, reimplementing another with different information available). Are there no better baselines?

W3.

- The training time seems to be a big disadvantage for this method, given a big assumption that the adversary can use any public images.
- The specific architectural choices regarding MI-ControlNet are not strongly motivated and explained.

W4. Others:
As of now, writing is not easy to follow, figures and tables are in a very different places from where they are referenced and it is not easy to understand how specific components fit into a bigger picture. It was necessary to put an effort to understand the big picture and filter-out many specific issues of lesser importance discussed along the most important ones.

**Questions:**

- weaknesses/concerns
- Could the authors tell me more about the target model and number of classes used in presented experiments?
- The authors could think on how to present their work from the big-picture to details. Even Fig. 1 is far too complicated to introduce proposed approach to a reader.

---

### Official Review · Reviewer_SpcD · 2025-11-01

**Soundness:** 3
**Presentation:** 2
**Contribution:** 2
**Rating:** 4
**Confidence:** 5

**Summary:**

This paper introduces a novel approach to black-box model inversion attacks (MIAs) by leveraging pre-trained latent diffusion models (LDMs) through an adapter network called MI-ControlNet. Unlike previous methods, the authors leverage diffusion model for MIA and achieve state-of-the-art performance.

**Strengths:**

- This is the first paper to leverage LDM for model inversion attack.
- The integration of ControlNet for MIA is novel.
- The proposed method achieve the state-of-the-art performance around numerous experiments.

**Weaknesses:**

- The most significant weakness of this paper is the complete absence of any discussion or analysis regarding query efficiency. For a black-box attack, the number of queries to the target model is a fundamental performance metric and practical constraint.
- While the results are impressive, the experiments do not fully isolate the contribution of the novel MI-ControlNet architecture from the inherent power of the pre-trained LDM. The paper is missing crucial baseline comparisons that would help quantify the value of their specific design. For instance, comparing MI-ControlNet against a more standard approach would provide a clearer picture of why their method is so effective. Without this, it's hard to distinguish the gains from the adapter's architecture versus the gains from simply using a strong generative prior like Stable Diffusion.
- (minor) The way the main figure is drawn makes it unclear what the key points are.

**Questions:**

See Weakness

---

### Note · Authors · 2025-12-10

I have read and agree with the venue's withdrawal policy on behalf of myself and my co-authors.